# A Review of Propagation and Restoration Techniques for American Beech and Their Current and Future Application in Mitigation of Beech Bark Disease

Andrea L. Myers [1], Andrew J. Storer [1], Yvette L. Dickinson [1,2] and Tara L. Bal [1,*]

1 College of Forest Resources and Environmental Science, Michigan Technological University, 1400 Townsend Dr, Houghton, MI 49931, USA; almyers@mtu.edu (A.L.M.); storer@mtu.edu (A.J.S.); yvette.dickinson@scionresearch.com (Y.L.D.)
2 Scion Research, Titokorangi Drive, Private Bag 3020, Rotorua 3046, New Zealand
* Correspondence: tlbal@mtu.edu

**Abstract:** The American beech (*Fagus grandifolia* Ehrh.) has been impacted by the beech bark disease (BBD) complex throughout the northeastern United States for over 100 years, but the disease has been present in the Great Lakes region only for around 20 years, requiring acknowledgement of the evolving context surrounding *F. grandifolia*. This disease threatens to remove a foundational tree species which is especially important ecologically for wildlife habitat and mast, and as a climax successional species. We review advances in propagation techniques of *F. grandifolia* with the goal of addressing their use in the rehabilitative restoration of forests affected by BBD. Natural regeneration and artificial methods of propagation are addressed, along with how they may be applied for mitigation. Silvicultural interventions are discussed that may be necessary to protect and release resistant seedlings to promote persistence. An existing framework is used to explore context necessary for decision making in restoration. Nucleated seed orchards of resistant trees may currently be the most effective and practical method for introduction of BBD-resistant *F. grandifolia* into affected northern hardwood forests.

**Keywords:** grafting; restoration; tree breeding; invasive species; *Neonectria*; *Cryptococcus fagisuga*

## 1. Introduction

Emergent diseases present challenges to modern forests and are becoming increasingly common and severe in response to global change, with land managers relying on a constantly evolving suite of forest health management tools [1,2]. Forest management techniques are used in an effort to protect forests that are faced with health challenges, but these efforts are often costly, difficult to implement, and sometimes ineffective or unsustainable. Sustainable forest management practices help support sustainable forests. These in turn provide important ecosystem services in many forms, from water quality protection to wildlife refuge and biodiversity havens.

One ecologically important tree which is threatened by an invasive disease complex is the American beech (*Fagus grandifolia* Ehrh.), which has been impacted by beech bark disease (BBD) in the northeastern United States for over a century. *F. grandifolia* is an ecologically important species in the northern hardwood forests of the United States and throughout its range in the eastern United States. It serves as a major source of forage as the nuts are consumed by a diverse suite of wildlife. *F. grandifolia* is also ecologically important as a late successional, shade-tolerant "climax" species in the northern Great Lakes region of the United States [3]. In this area, sugar maple (*Acer saccharum* Marsh.) and Eastern hemlock (*Tsuga canadensis* L.) fill similar successional roles, but neither creates hard mast that can compete with *F. grandifolia* for quantity and quality [4]. Beechnuts serve as forage

for mammals at many trophic levels, and increased beechnut crops are associated with increased predatory animal populations (e.g., [5,6]).

There are seven genera in the family Fagaceae (*Castanea*, *Castanopsis*, *Chrysolepsis*, *Fagus*, *Lithocarpus*, *Quercus*, and *Trigonobalanus*). There are 11 species within *Fagus*, with only *F. grandifolia* occurring naturally in North America [7]. The range of *F. grandifolia* extends from Cape Breton Island, Nova Scotia, Canada, to the gulf coast of the United States, and from the coast of the Atlantic to parts of Wisconsin, Illinois, Missouri, Alabama, and into Texas [8]. Within this range, the tree can occur on a variety of soils, primarily on gray-brown podzolic or laterite soils, although it can be found on other soils including limestone in some areas [8]. The wide range of *F. grandifolia* is possible as it can withstand variable environmental conditions, occurring in areas with precipitation ranges from 580 mm to 1270 mm, temperature ranges from −42 °C to +38 °C, and areas with growing seasons from 92 to 280 days [8]. It is wind-pollinated, and thus no pollinators are necessary. The hard wood of *F. grandifolia* can be utilized as fuel wood, lumber or veneer logs, railroad ties, and pulpwood, among other uses [9].

Technological and methodological advances have increased the success rates of propagation of *F. grandifolia* over traditional methods, increasing the suite of tools available for the mitigation of BBD, which often occurs through the creation and breeding of young trees resistant to the disease complex. Renewed attention is being placed on BBD mitigation (where the impacts and spread of a disease are reduced as much as possible with the understanding that total elimination of a disease or disease agent is impossible) as the scale insect expands its range into the western and southern range extents of *F. grandifolia* [10,11]. In order to present the state of techniques available at this time, we have performed a literature search for the propagation, micropropagation, restoration, and silviculture of *F. grandifolia* in North America. We placed emphasis on finding peer-reviewed studies with reproducible methods for the propagation or restoration of *F. grandifolia*, silviculture in BBD affected stands, and control of BBD and the individual biotic components. When peer-reviewed literature was not available, we include grey literature, such as handouts from extension offices or websites from reputable sources, such as arboretums or state agencies. Our goal is to present a comprehensive picture of the state of current propagation methods for practitioners interested in the sustainable conservation of this species.

Retaining genetically diverse, healthy *F. grandifolia* in the landscape is important as BBD is not the only forest health issue exerting pressure on the species. Beech leaf disease (BLD) is an emergent disease first described in the United States in Ohio [12–16]. Because it is very novel (first described in 2020), we are still understanding the ecology of BLD, a leaf disease of *F. grandifolia* which causes interveinal thickening and chlorosis of leaves after trees are infested by the nematode *Litylenchus crenatae mccannii* (Anguinata) [13]. The beech leaf mining weevil (*Orchestes fagi* Linnaeus) is an introduced pest that has been present in Nova Scotia, Canada for at least 15 years [14]. The impacts of this invasive defoliating pest can at times compound the effects of BBD to increase mortality from the disease, but generally are associated with increased canopy decline and mortality of *F. grandifolia* [14]. Climate change is also expected to reduce the southern range extent of *F. grandifolia* [15]. Other unknown challenges are likely to emerge for *F. grandifolia* and other northern hardwood species in the future, so retaining resilient, healthy populations will be critical for ecosystem health.

Beech bark disease has been present in the northeastern region of the United States for over a century but was only described in Michigan (the western edge of the disease front) in 2001 [16]. Because of the longer history in the northeast, much of the fundamental literature about BBD has originated east of the Appalachian Mountains [17,18], and thus there is a need to carefully evaluate management decisions made for BBD mitigation as it enters new regions or forest types. Fundamental knowledge concerning the conservation of *F. grandifolia* has been identified as an area of concern [19] because fundamental knowledge of *Fagus sylvatica* L., the European beech, is often substituted, or information for the genus *Fagus* is used. One example is the *Woody Plant Seed Manual* [20], an extensive manual

for the proper storage and planting of woody plant seeds published by the United States Forest Service. This manual provides more extensive information for the seed treatments of *F. sylvatica* due to availability of data, despite *F. grandifolia* being the only *Fagus* species native to North America.

The objective of this review is to summarize current efforts to propagate and restore *F. grandifolia* as an ecologically important species and evaluate the relative benefits and drawbacks of different methods in their applications for restoration, particularly in the interest of long-term, sustainable restoration. Reviews have been completed detailing the ecology and impacts of BBD [18,21] and habitat information about *F. grandifolia* is available [8,20]. Therefore, we focus on information relevant to propagating *F. grandifolia* and restoration techniques which may be used to conserve and propagate BBD-resistant trees. This information presented together provides a foundation which can be used to develop management recommendations to sustain *F. grandifolia* as a forest component. We will not go into the current valuable work being done in tree improvement, though propagation techniques are often necessary for the creation of effective tree breeding and improvement programs. This work is invaluable for the future quality of the species in question, but it is outside the scope of pure vegetative propagation. Here, we explore management for the mitigation of BBD and restoration of *F. grandifolia* and discuss (1) the impacts of BBD on regeneration, (2) methods of management utilizing natural *F. grandifolia* regeneration and identification of naturally BBD-resistant trees in the landscape, (3) techniques currently being utilized to propagate BBD-resistant *F. grandifolia*, and (4) how these techniques can be applied towards *F. grandifolia* restoration.

### 1.1. F. grandifolia Ecology and Natural Reproduction

*F. grandifolia* reproduces prodigiously through seed and vegetatively via root suckering [8,22]. Suckers are produced from reparative root buds formed in response to senescence or injury [23]. Dense thickets of regeneration can establish around injured or dying *F. grandifolia* trees [24]. It is even possible for sucker-originated young trees to outcompete seed-originated young trees to comprise the majority of regeneration in patches [25]. Wagner et al. [26] further describes the mechanics of regeneration characteristics within the genus *Fagus*.

When considering the value of *F. grandifolia* in eastern North American forests, its use in wood products is sometimes overlooked or discounted. In the northeast, *F. grandifolia* regeneration was historically undesirable, and considered a nuisance preventing the establishment of other species [27]. However, the wood of *F. grandifolia* is being explored as an option for flooring timber in Canada [28] and trees felled in salvage cuttings in response to BBD are utilized for pulp, rail ties, or fuel wood where possible. BBD can cause sunken lesions, a type of timber defect, thereby reducing the economic value of timber forests affected by BBD [29]. *F. grandifolia* is good-quality firewood, ranking below hickory (*Carya* spp) and white oak (*Quercus alba*) in heating value [9].

Economic value aside, *F. grandifolia* also holds great importance throughout its range as a foundation species. Many birds prefer large *F. grandifolia* stems as a foraging substrate, including the pileated woodpecker (*Dryocopus pileatus*), Acadian flycatcher (*Empidonax virescens*), and scarlet tanager (*Piranga olivacea*) [30]. *F. grandifolia* trees and snags offer cavities in abundance across a range of forest types, creating appropriate nesting sites for yellow-bellied sapsuckers (*Sphyrapicus varius*), southern flying squirrels (*Glaucomys volans*), and wood ducks (*Aix sponsa*) [31,32]. *F. grandifolia* mast is a critical component in many forest food webs. It serves as forage for mammals at many trophic levels [5,33]. These specific roles cannot be filled by other species with similar life histories, so *F. grandifolia* is a foundational forest species.

### 1.2. Beech Bark Disease Pathosystem and Ecology

In BBD, a scale insect infests *F. grandifolia* trees and creates feeding damage sites on the bark, which are subsequently infected by a fungus from the genus *Neonectria*, causing

decline and eventual death [16]. The insect component of the disease is usually the non-native *Cryprococcus fagisuga* Lind. (Hemiptera: Eriococcidae), the felted beech scale or beech scale, a wingless, parthenogenetic insect with piercing–sucking mouthparts which it uses to feed on the phloem of trees, creating the infection courts necessary for the fungal pathogen to enter the tree [34,35]. One single healthy adult scale insect arriving on a tree may be enough to fully infest that tree because it can reproduce clonally until the tree is fully infested [35]. The two commonly identified fungal species associated with the disease complex are *Neonectria faginata* Castlebury and *Neonectria ditissima* Samuels and Rossman [36]. Both species of fungi apparently cause mortality through the same mechanism: slow annual accumulation of necrotic cankers on the above-ground portion of the tree as the tree is slowly girdled to death.

As this disease progresses, a number of symptoms occur. Necrotic cankers appear with the fungal infections, which increase in size annually through reinfection. Trees exhibit reduced growth which declines with increasing severity of symptoms and internal defects [37]. Occasionally sunken lesions occur. Tarry spots may occur on trees, where dark exudate appears on the bole. The canopy declines as the infection increases around the tree. Eventually the infection girdles and kills the tree [10,16], or the tree may succumb to additional stressors such as wind snap or other fungal diseases.

Some facets of BBD have become better understood over time. One example of our change in understanding is the discovery that over time, the dominant fungal component of the disease complex shifts from high levels of *N. ditissima* to *N. faginata* [36,38]. While the mortality associated with BBD was first described as only rapid mortality, it is now understood that BBD may be a rapid or a slow decline of overstory trees via girdling, up to 30 years to kill individual trees [38]. Generally, a strong suckering response is expected to occur in the understory [39], but recent literature from the expanded range of the disease has described differing regeneration reactions, indicating that suckering is not a direct response to BBD stress alone, but to the stress from the disease coupled with other environmental factors such as damage associated with harvest activities [40]. Continued research on BBD is still necessary both to fill gaps in fundamental knowledge, and to describe novel dynamics as the range of the disease increases.

### 1.3. Direct Control of BBD

Few direct controls are recommended for BBD, and the methods that are described are primarily only for retention of individuals, such as landscape trees. This is partially due to the fact that, although control of the scale insect is possible (but difficult and expensive), *Neonectria* fungi are ubiquitous in North American forests, and so cannot be removed entirely from the system where *F. grandifolia* occurs. In addition, shortly after it has arrived in an area, scale infestations become ubiquitous due to the mobile and wind-dispersed first instars that readily disperse [41,42]. The facts that *Neonectria* cannot functionally be removed, and that scale insects disperse widely after arrival and reproduce clonally, makes individual tree control costly and time- and labor-intensive due to the need for repeated, specialized control.

Biological control has been explored for BBD, with a number of predators identified as feeding on beech scale, though none have been described as effective in the broader control of BBD [43,44]. The twice-stabbed lady beetle (*Chilochorus stigma* Say) dispersed ineffectively and did not feed on all stages of the scale insect [44]. A velvet mite (*Allothrombium mitchelli* Davis) has also been noted, but little is known about the species [43].

Chemical controls have been used for the scale. Though insecticidal soap, horticultural oils, and bark spray insecticides can be used, treatments must be repeated. Scale insects are minute enough to escape into bark crevices, making bark sprays ineffective. Chemical controls are only recommended for individual ornamental trees [45]. *Neonectria* fungus has been controlled in apple species (Rosaceae) with copper oxychloride and copper oxide, and apple-specific fungal control compounds, but no reports of the fungicide efficacy on *Fagus* species are available [46,47]. Prophylactic control of *Neonectria* (where the control is

enacted to prevent the fungus from physically reaching the tree prior to infection occurring) is also difficult due to the widespread, rain- and wind-dispersed nature of the fungus.

Physical controls do work well to eliminate scale in individual trees, such as ornamental landscape trees. Horticultural oils can be applied during the dormant season; however, the mobile nymphs (the ideal target) emerge in August or September, reducing efficacy unless oils are applied with precise timing [48]. Scales may be physically washed or brushed from the boles of trees [10]; however, a single adult scale remaining is enough to re-infest the tree. A combination of physical and biological control, consisting of field paint containing a strain of *Bacillus subtilis* Ehrenberg coated over active cankers, reduced spore release from apple trees (*Malus spp* Mill), but the entire canker must be painted shortly before spore release [49]. In forests where *Neonectria* is endemic on other species, such as in beech–maple forests, this method is impractical.

Few cultural controls exist, and the few that do exist are focused on removal of BBD-susceptible *F. grandifolia* [50]. Cutting of both susceptible overstory trees and susceptible regeneration followed by herbicide application removes and prevents further establishment of disease-susceptible tree regeneration [51]. This method can be coupled with retention of disease-resistant overstory trees to remove advanced regeneration which is disease-susceptible and allow new, potentially resistant regeneration [52]. There is evidence that removal of diseased *F. grandifolia* and the retention of disease-free trees can improve the genetic quality of *F. grandifolia* stands over long periods of time [53]. Because direct control methods to affect the scale insects or fungus have not been adequately developed for long-term use, cultural controls are the typical recommended control action for BBD in forests.

## 2. Natural Vegetative Propagation of BBD-Resistant *F. grandifolia*

Some *F. grandifolia* are resistant to BBD, and this small portion of trees are the focus of most propagation efforts. About 1 to 3% of *F. grandifolia* trees are truly resistant to the insect portion of the disease complex [48,54]. Even when challenged in the field with scale egg inoculations, these trees display resistance to establishment of the scale insect [48,55]. Research is ongoing to determine the cause of the resistance, but bark protein profiles are different in susceptible and resistant trees [56]. One of the earliest studies exploring the mechanism of scale resistance proposed a gene which encodes a metallothionein-like protein as a candidate for conferring resistance, but no quantitative studies have yet been performed to support this hypothesis [34]. Currently, a multi-agency project is exploring the genetic basis for resistance in-depth [57].

A potential cultural control is enhancing the proportion of seed being released by resistant trees that are not susceptible to the scale insect, in conjunction with other control measures such as the silvicultural treatments. Resistant *F. grandifolia* trees can be retained in the landscape as potential seed trees, allowing propagules from these resistant trees to reenter areas where *F. grandifolia* is being removed by BBD. In Wisconsin and Michigan, for example, agencies have made information available to the public describing how to identify resistant *F. grandifolia* and encouraging their retention as seed trees [58,59]. Landowners are advised to watch for trees which are not infested by scale, and if such trees are found, to retain them and if possible, implement a sanitation cut around them. Over time, these long-lived trees should survive longer than trees which are susceptible to disease and become the most dominant *F. grandifolia* trees in areas affected by BBD, and thereby the most successful reproducers as susceptible trees succumb to the disease before reaching the dominant position in the canopy.

Resistant trees may be identified in the landscape by the lack of white-washed appearance due to scale. Scale infestation can be identified by the presence of white waxy chaff on the bole of the tree. If no disease signs (scale signs, cankers) are visible on the tree after careful visual inspection of the entire bole with the naked eye and binoculars, the tree is considered potentially resistant. Potentially resistant trees should be retained and re-inspected over successive years. Other visible symptoms of BBD include the presence of annually increasing necrotic cankers, or less commonly tarry spots or sunken lesions [16,17].

These symptoms only develop after the tree has been infected by the fungal component of the disease. The absence of scale infestation indicates potential resistance of the tree, as some lesions and other signs of decay can occur on *F. grandifolia* trees that do not develop BBD and can be difficult to distinguish from BBD symptoms without experience. Potentially resistant trees identified in the field can be confirmed through field challenge tests [48,55]. Field challenge tests should only be performed in areas where scale has already arrived and established to prevent introduction of the disease to new areas.

A potential source of resistant young trees may be transplanting of known-parentage, resistant seedlings or suckers, though *F. grandifolia* has a reputation as difficult to transplant [60,61]. The transplanting of root suckers for restoration has been successfully applied in the reforestation of other landscapes, notably in tropical hardwood species and arid landscapes (e.g., [62,63]). Propagation of *F. grandifolia* by transplanted root suckers is not widely reported in scientific literature but is a known route for many trees. As the application of root sucker propagation for restoration of other species can be successful, it should be explored for *F. grandifolia*, which so readily root suckers.

## 3. Artificial Vegetative Propagation of BBD-Resistant *F. grandifolia*

### 3.1. Micropropagation

Where micropropagation methods have been properly refined, they can be used to produce a large number of clonal plantlets in vitro in a short amount of time. Micropropagation techniques can be broadly classified into two methods: organogenesis, where many shoots are cultivated from tissue of an existing tree which then form roots, and somatic embryogenesis, where many fully formed embryos are cultivated from tissue of an existing tree, which contain root and shoot embryonic structures. Micropropagation allows for short timelines for creation of a large number of clonal plantlets, but not all produced clones will recruit into soil media [64]. Still, because of the volume of clones which can be produced at the same time, it would be ideal to define methods for micropropagation, because they would allow for the largest volume of clones to be produced in short rotations.

Methods exist to create plantlets through various micropropagation methods, but methods for successful transfer to soil are not well defined. Beech (*Fagus spp.*) plantlets can be produced through organogenesis [65,66] but no methods have been published to reliably transfer them to soil. Beech embryos can be created through embryogenesis, but again no process has been developed for successful transfer [67]. Micropropagation methods for *F. grandifolia* have been explored [65,68], though no method of propagating and growing plants through micropropagation has proven reliably successful. Continued research is warranted in these methods, as development of procedures to transplant these plants to soil would allow for production of a large number of clones of resistant trees.

### 3.2. Grafting

Grafting can create clonal plants which are already established in soil. Handling time for individual plants is generally longer compared to micropropagation, but overall success rates are higher. Grafting is currently the leading method for clonal propagation of *F. grandifolia*. Scions may be harvested from the outer canopy of a vigorous, BBD-resistant *F. grandifolia* in the spring before flush occurs (usually February to March). *F. grandifolia* grafting uses freshly cut scions stored for no more than two weeks before grafting [69,70]. Large scions (up to 2 m in length) may be collected and trimmed to size immediately before grafting.

Bench grafting methods produce *F. grandifolia* with published success rates as high as 30% [69]. In bench grafting, a scion from a resistant tree is joined with a containerized rootstock from the same species and allowed to heal fully in a greenhouse before planting. Generally, top cleft grafts are used because tools are available to standardize the cut, minimizing technician error. Top cleft grafts require a very precise diameter match of the scion and rootstock. If an exact diameter match is not possible, an apical veneer graft or two staggered veneer grafts may be used, or a modified side graft in the case of small diameter

rootstock [69,70]. Apical veneer grafts closely resemble side veneer grafts, but all rootstock above the graft union is removed.

The application of hot callus grafting has achieved average success rates as high as 57% when applied by experienced grafters [70]. In hot callus grafting, top cleft or apical veneer grafts are used to join scions to dormant rootstocks (stored at 4–6 °C and lightly watered). After trees are grafted, they are moved to a cold chamber where ambient air temperatures remain cool (4–6 °C) and only the graft union area is kept inside an insulated heated space (24 °C). Trees are carefully checked for new callus formation at the graft union site starting at about three to four weeks after grafting. When callus tissue has formed, the trees are moved out of the hot callus apparatus to a standard greenhouse or shadehouse [70].

In all grafting, tools, scions, and rootstocks should be disinfected with a 70–80% ethanol solution to prevent contamination. After cutting, the cambium of the rootstock and scion should be laid fully flush together, matched exactly on at least one side of the cut surface. If an exact match is not possible, a larger scion should be selected and matched on one side [69]. Grafts should be gently, yet firmly secured with flexible materials such as grafting rubbers. Rubbers should be wrapped with space sufficient for callus expansion (Figure 1). The entire tree should be dipped in warm (55 °C) paraffin wax past the graft union to prevent drying.

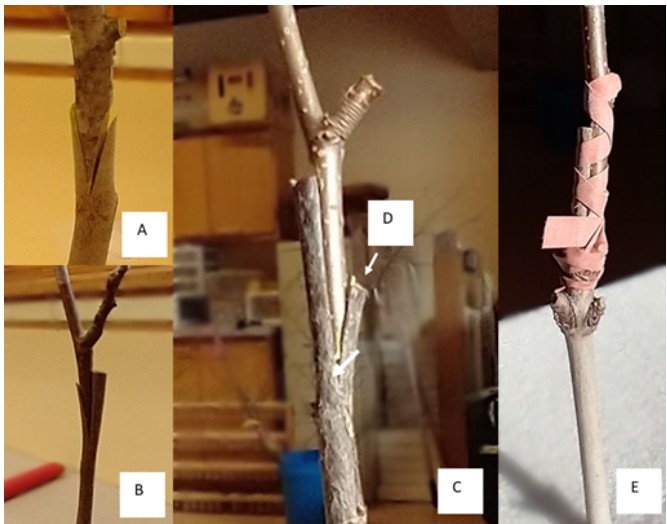

**Figure 1.** Grafts acceptable for use in *F. grandifolia*. (**A**) Top Cleft graft, performed with a FieldCraft Top Grafter. (**B**) Apical veneer graft. Two veneer grafts may be staggered on opposite sides of the rootstock. (**C**) Modified side graft with (**D**) enhanced sap drawer. (**E**) A graft union wrapped with space sufficient to allow callus formation.

Grafting is the currently accepted method for propagation of BBD-resistant *F. grandifolia* because methods exist for successful propagation resulting in trees suitable for planting, but there are some drawbacks to the method. Unlike micropropagation, which creates a whole clonal plantlet, grafting produces a tree which is resistant to BBD above the graft union but is still likely susceptible to BBD below the graft union. Graft unions can fail spontaneously years after healing. While success rates are high and methods have been refined for hot callus grafting, space and monetary investments reduce the accessibility of the method. Hot callus grafting requires cold storage and the construction of special hot chambers, limiting space for plants, resulting in low numbers produced at one time. Traditional grafting techniques (those without the aid of a specialized tool) require practice to optimize yield. The relative difficulty to propagate *F. grandifolia* may lead to a shortage of rootstock availability in private nurseries.

Grafted trees are usually placed in seed orchards to allow long-term maintenance of grafted trees. The graft union will remain fragile for an extended period of time, so the trees should be monitored until they are robust enough to resist damage at the union site. Seed

orchards allow production of high volumes of seed of known parentage by limiting pollen sources to the individuals within the orchard (locations are typically in areas with few to no external pollen sources). While this limited breeding stock is desirable for creating crosses of two known resistant parents, it comes at the cost of an inherent genetic bottleneck if sufficient genetically diverse parent trees are unavailable.

A well-designed seed orchard can effectively capture all or a very large proportion of the genetic diversity in a population, but in situations where genetic diversity is limited in the wild, such as in the case of BBD, where only 1–3% of the population will serve as suitable parent trees, populations may be inherently genetically narrow. A well-designed orchard can artificially increase available genetic diversity by design through the intentional crossing of two resistant trees that would not have the ability to cross in the wild due to distance between the individuals.

## 4. Applications in Restoration

Generally, the technology of a restoration project will be dictated by constraints on time, facilities, and money. In the case of *F. grandifolia*, there is still fundamental knowledge missing about the species in regards to seed propagation and artificial regeneration [19], so restoration activities should be planned understanding that emerging knowledge could change the context surrounding planned restoration activities.

In complex problem solving, conceptual frameworks can help guide the creation of a focused plan of action. We have applied the conceptual framework that was developed by Jacobs et al. [71] for the restoration of American chestnut (*Castanea dentata* (Marsh.) Borkh). The authors suggest that to create an effective restoration plan, definition of goals, available inputs and limitations should be considered within the context of ecology, society, and technology spheres [71].

Societal context describes public perception of the species and program, governmental policies and regulations in the area for restoration, and relationships between agencies working on the disease. Ecological context should include background information on the species, as well as an accurate snapshot of the ecology of the area targeted for restoration. The level of degradation should be assessed individually for the target area for each restoration project. Accurate, timely information about the targeted area can identify ecological barriers to restoration if they exist. Technological context describes the techniques necessary for reintroduction of a species. Generally, the technology of a restoration project will be dictated by constraints on time, facilities, and money. In the case of *F. grandifolia*, there is still much fundamental propagation knowledge missing, so restoration should be planned with the understanding that emerging knowledge could change the context surrounding activities.

Propagation can be used to develop resistant trees, but sustainable, effective restoration requires the planting and continued survival of these trees in the field. When identifying goals for restoration of BBD-affected forests, it is important to preserve any desirable overstory that should not be disturbed (both a small number of resistant *F. grandifolia* and the remaining co-occurring species), so transformative restoration techniques are likely appropriate in these areas. Rehabilitative restoration, in which forests are restored to a state similar to preexisting conditions, but possibly to a different or degraded state still, is appropriate where forests have been degraded but not eliminated. Forests can be converted or transformed as part of rehabilitative restoration. In conversion, the forest overstory is removed entirely or partially, and a new forest is grown on the site. Transformative restoration involves gradual removals and replacement of portions of the overstory (Figure 2) [72].

In transformation, partial removal of competing vegetation creates availability of growing resources, such as light, water, and soil nutrients for the newly planted individuals of the target species. In BBD-affected forests, a combination of techniques can be used to accomplish the goals of transformative restoration. Cutting and removal of existing vegetation may be necessary to remove both diseased and dying overstory *F. grandifolia* trees and non-resistant *F. grandifolia* regeneration, as well as other competing understory

vegetation (e.g., invasive grasses, shrub species [26]) before restoration plantings can occur in BBD-affected forests. The silvics for *F. grandifolia* are known, so thorough surveying enables site selection that is likely to provide the growing space and resources necessary for success of young plants. Information on the severity of BBD and scale infestations can inform decisions for where to focus restoration efforts. Selecting a site with high mortality and low regeneration, if possible, could eliminate the need for site preparation. If these sites are not available, removal of overstory diseased *F. grandifolia*, coarse woody debris, or regeneration may be necessary. Mechanical and chemical control of competing vegetation as described by Ostrofsky and McCormack [50] would be appropriate site preparation.

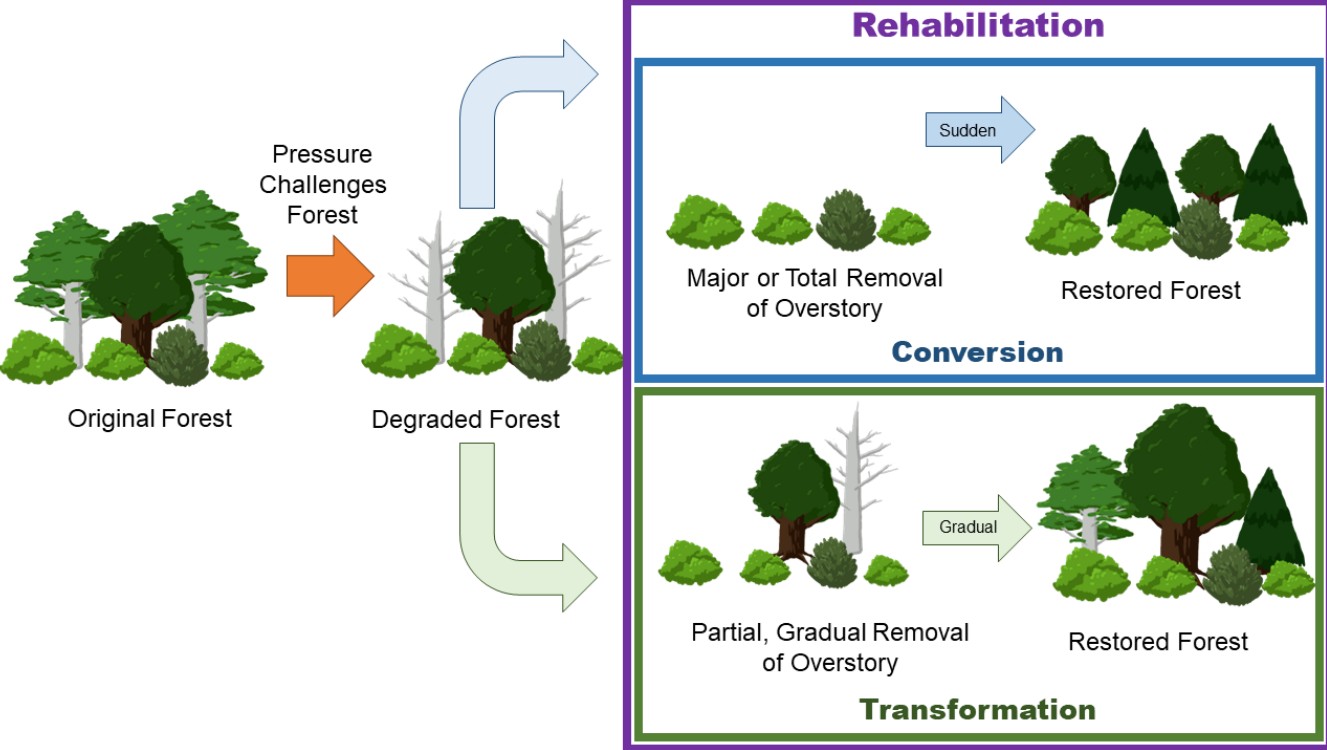

**Figure 2.** Types of rehabilitative restoration. Conversion focuses on the total or major removal of overstory species and replacement with other appropriate species. Transformation emphasizes the gradual or partial removal of species and replacement with a suite of species. Rehabilitative restoration may restore some preexisting species in the degraded forest, but the goal is not to recreate the pre-disturbance condition. Rather, this technique accepts a degraded or changed forest, and emphasis is placed on restoring function.

Removing existing vegetation may be necessary before plantings can occur but is only necessary in cases where robust regeneration is occurring (Figure 3). Sites without the strong "thicketing response" do exist in the western extent of the species, but this may be related to time since arrival of the disease [40] or other environmental or physical factors, such as aspects of the ground on which sprouts occur [22]. Ground surveying enables site selection that is likely to provide optimum growing space and resources necessary for the success of young plants. Knowing the severity of BBD and scale infestations can inform decisions regarding where to focus efforts. Selecting a site with high mortality and low regeneration could eliminate the need for intense site preparation.

*F. grandifolia* seedlings would likely be well suited to interplanting because of their tolerance of moderate to high shade in their early regeneration niche [8]. Interplanting, or planting seedlings among existing forest vegetation, has been utilized to enhance regeneration of other challenged species in the Fagaceae family, but careful site preparation is necessary to meet the light demands of these seedlings [73,74]. The same site preparations

may not be necessary if natural gaps due to BBD mortality can be utilized as planting sites. Research is necessary to quantify the amount of cover that would best support *Fagus* seedling growth in interplantings, and methods for the planting of larger seedlings or sowing of seeds in natural, degraded systems.

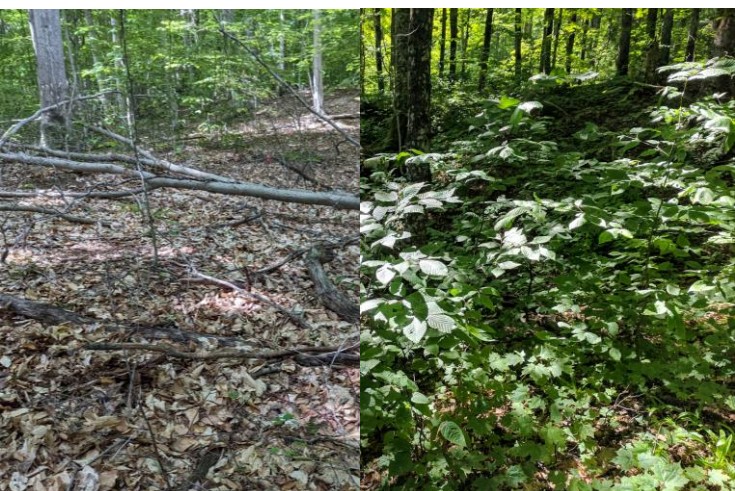

**Figure 3.** Sites within the same region with differing regeneration responses. Left, a site with above ground *F. grandifolia* mortality but no suckering response in the understory. Right, a site with above ground *F. grandifolia* mortality and an early thicketing response. Planting of trees would be possible with little control in the site with no suckering response, but intense silvicultural treatments would be necessary to plant in the site with a *F. grandifolia* thicket response.

With careful selection of parent trees, the highest proportion of genetic diversity can be retained [75]. Within the western range of *F. grandifolia*, if local provenance and genetics are preserved, the limited number of resistant parent trees occurring in the landscape will inherently limit the genetic diversity of seed orchards. When combined with a desire to retain yet-unknown genetic traits that may hold the key to combating future health challenges, the importance of propagating a genetically diverse population of parent trees increases. Careful consideration should be paid to selection of genetically diverse scion donor trees in the creation of local provenance orchards [76,77]. In traditional orchards, thorough selection of all available BBD-resistant parent trees will enable a large number of controlled genetic crosses. We recognize that the concept of locality can be dependent on artificial boundaries, so "local" may mean the extent of a park, a county, a state, etc., so locality must be defined by the agency pursuing the propagation, which will determine the number of parent trees needed for "sufficient" genetic diversity and complete representation of genetic profiles.

Robust young trees could be interplanted in areas where mortality has occurred, opening the canopy and freeing up both above and below ground resources and suitable microsites, to serve as nucleation centers for restoration (Figure 4). In nucleation, the species of interest is cluster planted, with the goal of drawing natural seed vectors toward the cluster, accelerating the pace at which the desired species spreads from the planting into the surrounding ecosystem [78]. While nucleation allows for immediate release of propagules in target forests, there is not the control over parentage found in traditional seed orchards. By interplanting resistant young trees within affected forests, uncontrolled pollination could occur between undiscovered resistant trees and known resistant crosses, allowing for persistence of resistant trees undiscovered by humans [79]. In *F. sylvatica*, inter-relatedness is high up to 40 m away from parent trees [80], and that distance is slightly lower in *F. grandifolia* at 20–30 m [81], but this is likely dictated by low seed dispersal distances. It is suggested that since *Fagus* is wind-pollinated, the pollen could travel long distances to introduce rare genotypes occasionally, with average ranges of pollen dispersal estimated from 40 to 180 m [81,82]. A potential solution then could be to move a potential resistant

seed source closer to resistant pollen sources. Open-pollinated *F. grandifolia* seedlings with only one resistant parent are about as susceptible to scale as seedlings with no resistant parents [83]; however, in planting resistant *F. grandifolia* in a patch near multiple known resistant overstory trees, it would increase the likelihood of resistant crosses occurring over time as large susceptible trees die and large resistant trees survive.

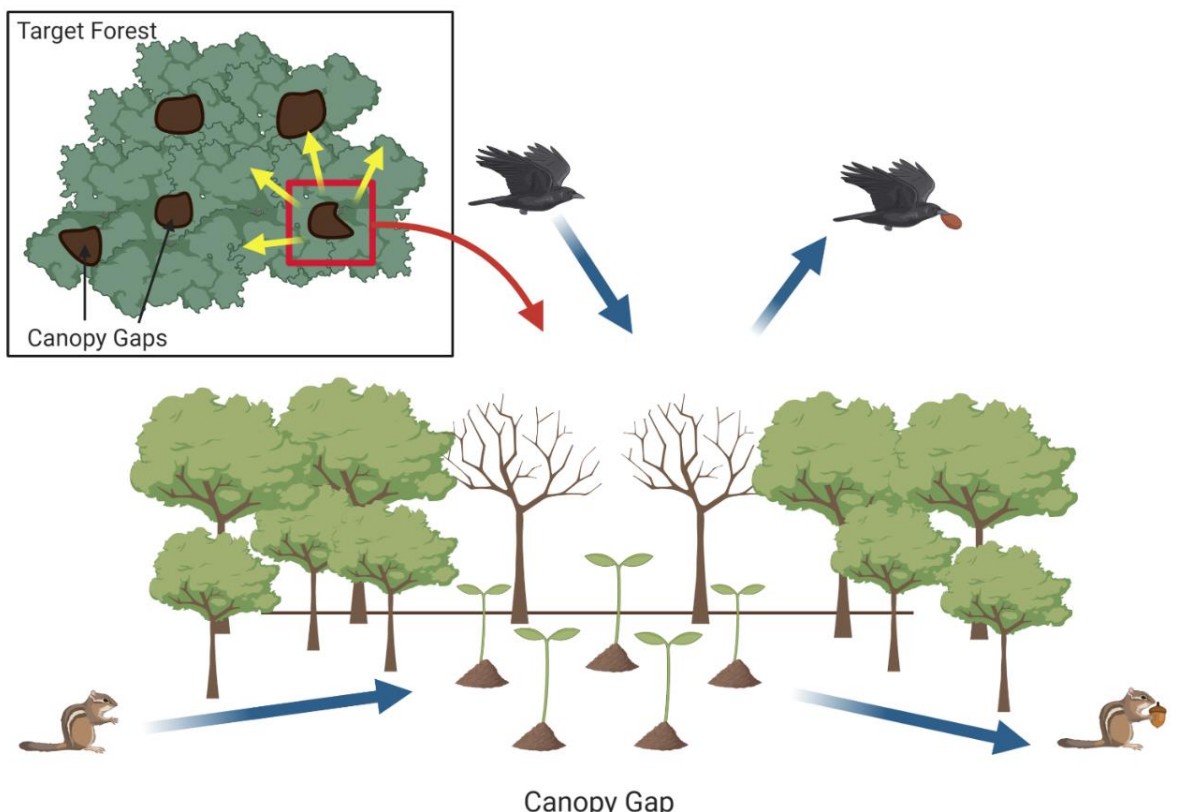

**Figure 4.** Nucleated *F. grandifolia* seed orchards are an option for restoration in targeted forests. A canopy gap created by beech bark disease is chosen as a nucleation site and resistant young trees are cluster-planted (competing plants in the understory must also be removed by mechanical methods prior to planting). Seed produced by the planted young trees attracts wildlife to the nucleated seed orchard. Wildlife aid in seed dispersal. In this way, nucleated seed orchards serve as a propagule for reintroduction into the entire target forest by natural seed dispersal pathways. Image created with Biorender.com.

Nucleation has largely been used in tropical forest restoration, and little literature exists examining efficacy in temperate forests [84]. Overall, it is more difficult to maintain the health of individual seedlings in nucleated plantings than in an orchard; however, each seed produced is released into an ecosystem in which it has a predefined niche, by the nature of planting the tree in disturbed areas which previously contained the species, making this a technique worth exploring as a sustainable restoration strategy for temperate forests. If grafted trees are planted directly into nucleation sites, they may be used to provide a relatively short-term pulse of propagules in affected forests as they will likely perish from stress before becoming dominant, but personal observation has revealed that grafted trees are reproductively mature as soon as they have healed from grafting when held in a greenhouse. A cluster planting of seedlings produced from bred resistant tree seed would provide a long-term injection of propagules but would take many years for the planted trees to reach reproductive maturity. The underplanting of resistant trees in nucleated seed orchards may serve as a restoration method that requires little active work in the form of sowing. Since it can be difficult to sustain long-term funding in forest restoration, this short-term pulse of propagules may serve as a sustainable alternative to

the expense of creating and maintaining a traditional orchard. Much research would be necessary to support this method, including quantifying life expectancy of interplanted *F. grandifolia* and grafted *F. grandifolia*, best methods for interplanting in degraded forests, and dynamics of reproduction in the western extent of *F. grandifolia*. This could serve as a stopgap measure in tandem with other efforts to propagate *F. grandifolia*, but the efficacy of these methods for *F. grandifolia* has yet to be studied in the field.

In Koch et al. [83] open-pollinated resistant seedlings (planted at a site where all susceptible trees were removed) were not significantly differently susceptible from a full-resistant cross improved planting. This suggests that planting resistant seedlings into an area with little competing *Fagus* regeneration could lead to improved levels of resistance if there is a source of resistant pollen nearby. When considering very diverse forest systems, nucleation is frequently much cheaper for restoring forest systems to an appropriate level of complexity compared to high-diversity plantation methods achieving similar restoration results, despite the high upfront costs of creating nucleated plantings [85]. Still, rate of spread and success vary across instances of application, with some evidence that small-seeded species gain the most benefit from nucleation plantings [86]. *F. grandifolia* may respond better to nucleation, since we have evidence that blue jays (*Cynocitta cristata* L.) are an important dispersal vector for the species [87]. Because birds are such an important dispersal vector for *F. grandifolia*, future research into nucleation plantings is warranted where natural dispersal of resistant trees is desired.

## 5. Management Recommendations

Restoration project decisions must be made on a contextualized, individual basis. In typical BBD-degraded forests, transformative restoration (where ecosystems are returned to a functioning ecosystem that is different from the historic ecosystem over an extended period of time through gradual replacement) with an emphasis on enrichment plantings (where the percentage of desirable species or genotypes is enhanced through interplanting in the target forest) and gradual removal of undesirable, susceptible trees (by cutting susceptible regeneration where it is interfering with desirable regeneration) to enhance the proportion of resistant genotypes in the landscape would be recommended. If cost and technology are limiting factors, simple preservation of resistant trees will retain a source of potentially resistant seed. If possible, hot callus grafting trees to create clones of resistant *F. grandifolia* allows for the most efficient creation of resistant trees currently. Tree breeding and improvement of the trees propagated through this method will increase resistance over time through improvement breeding. Programs are in operation currently in the Great Lakes region of the United States that will produce resistant-cross seed in the near future [88].

Agencies which desire resistant seed produced from a traditional orchard should prepare now, for the commonly practiced methods, while successful, will begin bearing seed at minimum 10 years in the future or more. Passive restoration techniques such as retention of seed trees are unlikely to restore *F. grandifolia* to its previous functional state, but have immediate benefits and can function as a reservoir of the species in the landscape. Because of the low relative effort and large potential reward, retention of seed trees should be strongly encouraged. Additional research to create new techniques between the demands of tree breeding and seed tree retention is needed to expand the suite of techniques for propagation of *F. grandifolia*, particularly as this species faces emergent challenges which threaten the sustainability of the species in natural areas.

Many gaps exist in our knowledge of propagating *F. grandifolia*, thus continued research on the methods described here could improve management techniques. Reliable transplanting measures should be determined to increase the success of restoration activities. While grafting rates have been improved with hot callus grafting, continued research could allow for a greater volume of plant material production, ideally through the successful transfer of micropropagated plantlets. Better understanding of the propagation of

*F. grandifolia* would benefit not only the mitigation of BBD, but improve the outlook for the species in the face of other emerging challenges.

**Author Contributions:** Conceptualization, A.J.S. and Y.L.D.; methodology, A.J.S. and Y.L.D. and A.L.M.; investigation, A.L.M.; resources, A.L.M.; writing—original draft preparation, A.L.M.; writing—review and editing, A.J.S., Y.L.D. and T.L.B.; visualization, A.L.M.; supervision, Y.L.D. and T.L.B.; project administration, Y.L.D. and T.L.B.; funding acquisition, A.J.S. and Y.L.D. All authors have read and agreed to the published version of the manuscript.

**Funding:** This research was funded by a Great Lakes-Northern Forest Cooperative Ecosystem Studies Unit (GLNF CESU) grant, Task Agreement Number P16AC01398, Beech Reintroduction at Pictured Rocks National Lakeshore and Sleeping Bear Dunes National Lakeshore, a cooperative agreement with the National Park Service.

**Institutional Review Board Statement:** Not applicable.

**Informed Consent Statement:** Not applicable.

**Data Availability Statement:** No new data were created in the writing of this manuscript. Data sharing is not applicable.

**Acknowledgments:** Bruce Leutscher was instrumental in creation of the initial funding proposal and *F. grandifolia* restoration project creation with the Department of Interior, National Park Service. The authors thank Scott Rogers at the Oconto River Seed Orchard, United States Forest Service for personal communications and training leading to increased understanding of the grafting and care of *F. grandifolia* seedlings. The images featured in Figure 2 were created by Andrea L. Myers in Adobe photoshop.

**Conflicts of Interest:** The authors declare no conflict of interest. Yvette L. Dickinson works for Scion research, registered as Forest Research Institute Limited administered by the New Zealand Crown Research Institute, a corporatized Crown entity. Funders had no role in the design of the study; in the collection, analyses, or interpretation of data; in the writing of the manuscript; or in the decision to publish the review.

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
