# Peer review of "A Review of Propagation and Restoration Techniques for American Beech and Their Current and Future Application in Mitigation of Beech Bark Disease"

_sustainability, doi:10.3390/su15097490_

Round 1

Reviewer 1 Report

The review manuscript on Propagation and Restoration Techniques for 2 American Beech and their Current and Future Application in 3 Mitigation of Beech Bark Disease is well organised and written. The manuscript can be accepted in the current form for publication. 

Author Response

Thank you for the review.

Reviewer 2 Report

Comments: A Review of Propagation and Restoration Techniques for American Beech and their Current and Future Application in Mitigation of Beech Bark Disease

The paper presents an interesting account of impacts of BBD on beech tree physiology and ecology. The paper addresses Natural regeneration and artificial methods of propagation, along with how they may be applied for mitigation. Silvicultural interventions are discussed that may be necessary to protect and release resistant seedlings to promote persistence. An existing framework is used to explore context necessary for decision making in restoration. Although introduction gives relevant background and challenges to Beech but the follwoing statement 'Beech leaf mining weevil (Orchestes fagi Linnaeus) is an introduced pest present in Nova Scotia, Canada for at least 15 years' as the context is not clear particularly by the word 'introduced'. Giving some insights into the economic significance of the beech would add further value in section 1.1. For me, the major drawback stems from the absence of a methodological overview of the study or at least the criteria for the selection of studies - both in terms of coverage and time - making it unclear as to what sort of studies have been focussed by the authors. Giving some background in the revised version would be enough to go ahead. Various propagation techniques discussed in section 2 and 3 are fascinating but it remains largely unclear as to what is their potential for opting them on a large scale or commercial-scale and their acceptability among farmers in the field instead of in the controlled conditions. And how far they have been used in open fields and what have been the success rates and uptake by producers also need to be considered. It would also be much desireable to give some account on the success of various in-vitro trials or experiments. 

Author Response

Thank you for these comments.  They are similar in line with other reviewers suggestions in line by line items.  For these specific comments:

Introduced is a less pejorative term for an invasive species. This is a standard use of the term within the field of ecology.  The examples of other pests mentioned in this paragraph is to highlight other known threats to American beech, making the case that retaining genetically diverse healthy beech will be important as other health issues exert pressure on the species as well.  

We have included a short explanation of the types of resources presented and how we decided on inclusions. There was no temporal limit- the disease was described in 1934, potential resistance was discovered in the 1980’s, development of silvicultural methods to improve components and propagation of the species is relatively recent, and the current state of propagation is the use of grafting and tree breeding.

Further information on the economic significance is unavailable. This tree is valued primarily as an ecologically important tree (or a nuisance tree in parts of its range), and is neither historically nor significantly economically important throughout much of its range, beyond the timber products listed.

Unfortunately, there are no applications of these methods in the field or for tree farmers. The methods listed are the only available routes for developed propagation at this time and it is extremely limiting. We are aware that this is an issue for American beech (Beckman, E.; Meyer, A.; Pivorunas, D.; Hoban, S.; Westwood, M. Conservation gap analysis of American beech. The Morton Arboretum: Lisle, IL, 2021) but there is no additional information to be presented for interested landowners or farmers at this time.

Reviewer 3 Report

Please modify the introduction as per the title.

·       In the introduction section “ One ecologically important tree which is threatened by an invasive disease complex is American beech (Fagus grandifolia Ehrh.), which has been impacted by beech bark disease (BBD), in the northeastern United States for over a century” please mentioned a total number of species in this genus and family. Average life of this tree? Geographical distribution of this species? commonly occurs in which type of soil?

·       Line 36-45: IS there any other species impacted by beech bark disease (BBD), in the northeastern United States? If yes, then please extend this paragraph by adding more relevant data. Please mention the ecological significance of this species as well. As in the first line the authors mentioned that this is an ecologically important tree that is threatened by an invasive disease.  

·       Line 46-43:  Along with BBD and BLD, would be better to add more about such diseases.

·       The species name should be in italics please check throughout the ms. For instance, see line 80.

·       Line 163-168: Direct control of BBD: Among the direct control biological, chemical, physical and cultural control, which one is better? Please provide a concluding sentence in this subheading.

·       Line: 197: The species name should be in italics.

·       Line 226-233: Please add more data about the symptoms of BBD, please follow te following paper https://doi.org/10.1016/j.foreco.2021.119753  

·       What are the environmental factors that affect the  Fagus grandifolia?

·       Would be better to provide the detail about the pollination strategies of this ecologically important species.

·       Please provide the history and spread of beech bark disease agents in a separate heading.

·       What are the effects of BBD on the growth of the host tree?

·       The figures are scientifically robust and sound-producing.

·       The Framework of the study is scientifically robust.

·       Please provide a conclusion under a separate heading and should be respond to the aims.

Author Response

Responses presented below each line item.

  1. In the introduction section “One ecologically important tree which is threatened by an invasive disease complex is American beech (Fagus grandifolia ), which has been impacted by beech bark disease (BBD), in the northeastern United States for over a century” please mentioned a total number of species in this genus and family. Average life of this tree? Geographical distribution of this species? commonly occurs in which type of soil

We have added information after line 37 to add information about the genus, distribution, soil demands, and economic information for the species.

  1. Line 36-45: IS there any other species impacted by beech bark disease (BBD), in the northeastern United States? If yes, then please extend this paragraph by adding more relevant data. Please mention the ecological significance of this species as well. As in the first line the authors mentioned that this is an ecologically important tree that is threatened by an invasive disease.

There are no other species affected by this disease in North America. This is reflected in the revision in response to reviewer one’s comment 1, that there are no other beech (Fagus) species in North America. Therefore, no additional edits have been made as the question is addressed previously.  Additional impacts would be due to the loss of beech as a foundational forest species but no other tree exhibits the disease.

  1. Line 46-43:  Along with BBD and BLD, would be better to add more about such diseases.

Included information about the symptoms and impacts of these issues.

  1. The species name should be in italics please check throughout the ms. For instance, see line 80.

This error was corrected at line 80 and line 197.

  1. Line 163-168: Direct control of BBD: Among the direct control biological, chemical, physical and cultural control, which one is better? Please provide a concluding sentence in this subheading.

We have added a recommendation for cultural controls at the end of this section.

  1. Line: 197: The species name should be in italics.

This error was corrected at line 80 and line 197.

  1. Line 226-233: Please add more data about the symptoms of BBD, please follow te following paper https://doi.org/10.1016/j.foreco.2021.119753

Symptoms were clearly listed in a new paragraph after line 138.

  1. What are the environmental factors that affect the  Fagus grandifolia?

This was addressed after line 37, along with the response to comment 2.

  1. Would be better to provide the detail about the pollination strategies of this ecologically important species.

This was addressed after line 37, along with the response to comment 2. This species is solely wind pollinated.

  1. Please provide the history and spread of beech bark disease agents in a separate heading.

We disagree with this as an inclusion. This disease is well documented in literature, and recent comprehensive reviews about the disease are available in the literature which we refer to. We believe that to include a history of the spread would be beyond the scope of this review. Specifically, we mention the following review as a comprehensive source for interested readers in line 109-110:

Stephanson, C. A.; Coe, N. R. Impacts of beech bark disease and climate change on American beech. Forests 2017, 8 (5), 155. DOI: 10.3390/f8050155.

Cale, J. A.; Garrison-Johnston, M. T.; Teale, S. A.; Castello, J. D. Beech bark disease in North America: Over a century of research revisited. For Ecol Manage 2017, 394, 86-103. DOI: 10.1016/j.foreco.2017.03.031.

And have cited throughout additional foundational literature and comprehensive reviews of many facets of this well-documented, long-standing disease:

Ehrlich, J. The beech bark disease, a Nectria disease of Fagus following Cryptococcus fagi (Baer.). Can J For Res 1934, 10, 593-692.

Houston, D. R. Beech bark disease: 1934 to 2004: What’s new since Ehrlich. In Beech Bark Disease: Proceedings of the Beech Bark Disease Symposium, Beech bark disease: Proceedings of the Beech Bark Disease Symp. Saranac Lake, NY, 16-18 June, 2004; Evans, C.

Wagner, S.; Collet, C.; Madsen, P.; Nakashizuka, T.; Nyland, R. D.; Sagheb-Talebi, K. Beech regeneration research: From ecological to silvicultural aspects. For Ecol Manage 2010, 259 (11), 2172-2182. DOI: 10.1016/j.foreco.2010.02.029.

Out of respect for the authors for these outstanding reviews, and in the interest of word count reduction, we have chosen not to summarize or paraphrase these studies, rather to introduce them to interested readers.

  1. What are the effects of BBD on the growth of the host tree?

Added information about growth rates to line 168-170, when discussing symptoms of the disease.

  1. The figures are scientifically robust and sound-producing.

Thank you.

  1. The Framework of the study is scientifically robust.

Thank you.

  1. Please provide a conclusion under a separate heading and should be respond to the aims.

The conclusions are included under section 5: Management Recommendations. This is a natural conclusion based on the scope of the paper, and it is not required under the review article structure to provide specifically a conclusions section. We feel that since there is no additional primary research presented, but present rather a synthesis of information, the management recommendations serve as a natural conclusion. We feel that this section would be most understandable and identifiable for professionals who search out this paper for application. We are willing to change this section heading, but would prefer to retain the “management recommendations” title.

Reviewer 4 Report

1.      The manuscript needs extensive revision of language and grammar throughout.

 2.      In case of multiple references cited in the text, I suggest to use single parenthesis instead of using separate parenthesis for each reference numbers. 

Please correct [1], [2] in Line-30 as [1,2]; [7], [8], [9], [10] in Line-48 as [7 – 10] and [54], [55], [17] in Line-165 as [17,54,55] etc.

The authors can also refer to the submission guidelines of the journal for the same and edit the manuscript accordingly.

 3.      The alignment of the entire manuscript needs to be ‘Justified’ so that the text appears evenly distributed between margins. 

4.      I believe it is preferable to incorporate the importance of the American beech tree as a fundamental forest species and its economic values in the introduction's first sections. (After L-36-38)

5.      L 33-35- Paraphrase the sentence

 6.      Please review and fix grammatical errors in the manuscript. I'll mention a few. L17, L 36- Use “that” instead of “which”, L 37- Add an article “the” before “American beech” in sentence

 7.      L 54- Be consistent throughout the document while using Beech Bark Disease/ beach bark disease or its abbreviation.

 8.      Paraphrase the sentence (L-110) to avoid repetition.

 9.      To minimize confusion, use either beech or the scientific name of the tree throughout the document.

 10.   L 336-339- Correct the grammatical error and simplify the sentence.

 11.   L 343-345- Why is there a reference at the end if this is a suggestion made by the authors of this manuscript? The sentence is not clear and meaningful-Please do paraphrase like this “The authors propose that in order to establish a successful restoration plan, goals, available inputs, and limits should be defined within the contexts of ecology, society, and technology”.

 12.   Put the sentence L 346-348 after L 351-352 to ensure logical flow.

 13.   Avoid repetition at L 336-337 and L 355

 14.   L 364-370- Please paraphrase the sentence to make it more meaningful. For eg: “Rehabilitative restoration, in which forests are restored to a state comparable to prior conditions but in a distinct and sometimes degraded one, is appropriate in instances where forests have been degraded but not eliminated. Forests can be altered or transformed as part of rehabilitative restoration. The forest overstory is removed entirely or partially during conversion, and a new forest is planted on the spot. Transformative restoration involves gradually removing and replacing sections of the overstory”.

15.   Check for the repetition.

“Beech nuts serve as forage for mammals at many trophic levels, and increased beech nut crops are associated with increased predatory animal populations” in Line-94,95 and “It serves as forage for mammals at many trophic levels, from myomorphic rodents to black bears (Ursus americanus), and increased beechnut crops are associated with increased predatory animals” in Lines 120-122.

16.   Check for the repetition

Cutting and removal of the existing vegetation may be necessary to remove both diseased and dying overstory F. grandifolia trees and non-resistant F. grandifolia regeneration, as well as other competing understory vegetation (e.g., invasive grasses, shrub specie” in Lines-382-385 and Removing existing vegetation may be necessary to cull both diseased and dying overstory F. grandifolia trees and non-resistant regeneration, as well as other competing understory vegetation in Lines-395-397.

17.   L 401-405- Check and fix grammatical errors and paraphrase the sentence.

18.   I suggest the authors to check and revise the manuscript for repetition of words, sentences etc.

19.   Try to avoid repeating the complete genus name of the binomial name if it has already been mentioned  (Ref L. 61).

20.   In sub heading 2 the binomial  name is not in italics, similarly in L.80.

21.   Some paragraphs do not convey any meaning relative to the context. Ref L.340-345.

22.   In figures, try to maintain the size of each component as the proportion  of the size of the components in a figure make a lot of sense (Fig 2).

23.   In the reference section, some references lack their year of publication (Ref No. 22), and some references do not have their year in bold format (Ref. No. 34). Reference number 15 seems confusing. Reference number 34 has a 'pp' at the end without indicating the page number. Some references use pp before the page number, some do not.

Author Response

Responses presented below each line item.

  1. The manuscript needs extensive revision of language and grammar throughout.

We believe the reviewer may be referring to stylistic standards, and have edited the manuscript to reflect a journalistic style, but would need additional guidance to move forward. The reviewer does not provide further detailed comments of revisions of language or grammar.  All authors on this manuscript are native, English is First Language, speakers.

  1. In case of multiple references cited in the text, I suggest to use single parenthesis instead of using separate parenthesis for each reference numbers.

We have changed the bracketed it text citations to a single bracket rather than multiple.

  1. Please correct [1], [2] in Line-30 as [1,2]; [7], [8], [9], [10] in Line-48 as [7 – 10] and [54], [55], [17] in Line-165 as [17,54,55] etc.

Please see response to comment 2.  Bracketed citations have been updated throughout.

  1. The authors can also refer to the submission guidelines of the journal for the same and edit the manuscript accordingly.

The citation style and format reflect the templates provided by the journal for use in Microsoft word.

  1. The alignment of the entire manuscript needs to be ‘Justified’ so that the text appears evenly distributed between margins.

The article was prepared in the provided journal template. Any major formatting changes such as spacing and justification would be editing during proof setting by an editor.

  1. I believe it is preferable to incorporate the importance of the American beech tree as a fundamental forest species and its economic values in the introduction's first sections. (After L-36-38)

Paragraph 113-124 has been moved to the beginning of the introduction, to introduce the importance of American beech at an earlier point in the manuscript.

  1. L 33-35- Paraphrase the sentence

We divided the sentence to improve readability.

  1. Please review and fix grammatical errors in the manuscript. I'll mention a few. L17, L 36- Use “that” instead of “which”, L 37- Add an article “the” before “American beech” in sentence

In standard English these examples are either incorrect or not necessary. In line 17, the following clause is non-defining, in which case “which” is appropriate. In example line 37, the definite article “the” is unnecessary for a singular proper noun and detracts readability, so we do not include it to reduce words and improve readability. We are willing to make changes if the journal provides a specific journalistic style to reflect.

  1. L 54- Be consistent throughout the document while using Beech Bark Disease/ beach bark disease or its abbreviation.

This capitalization has been formatted consistently throughout. We do not use acronyms when they are the first word in a sentence. If this is stylistically inappropriate for the journal, we can revert first words to acronyms throughout the text.

  1. Paraphrase the sentence (L-110) to avoid repetition.

We have improved readability of the sentence at 110.

  1. To minimize confusion, use either beech or the scientific name of the tree throughout the document.

All instances of the specific species name mentioned have been changed to F. grandifolia. Where beech occurs in the manuscript currently, it is where it is part of another word, or part of a name of a different species.

  1. L 336-339- Correct the grammatical error and simplify the sentence.

There is no grammatical error in the indicated sentence identifiable to edit.

  1. L 343-345- Why is there a reference at the end if this is a suggestion made by the authors of this manuscript? The sentence is not clear and meaningful-Please do paraphrase like this “The authors propose that in order to establish a successful restoration plan, goals, available inputs, and limits should be defined within the contexts of ecology, society, and technology”.

This is the specific framework that was applied in the following paragraphs and the reference is to the authors of Jacobs et al, which is also referenced in the sentence immediately prior to this sentence.  “The authors” refers to the authors of that manuscript, not “we authors”. The citation indicated presents a framework that defines needed informational input to choose restoration activities. This framework helps practitioners decide which information is important to begin the decision making process. This is a common practice in restoration ecology.

  1. Put the sentence L 346-348 after L 351-352 to ensure logical flow.

We have left the sentences 346-348 before the following paragraphs, to forewarn readers as they begin this section that the specific context described here should not be blanket applied to their project.

  1. Avoid repetition at L 336-337 and L 355

This sentenced has been shortened to reduce repetition.

  1. L 364-370- Please paraphrase the sentence to make it more meaningful. For eg: “Rehabilitative restoration, in which forests are restored to a state comparable to prior conditions but in a distinct and sometimes degraded one, is appropriate in instances where forests have been degraded but not eliminated. Forests can be altered or transformed as part of rehabilitative restoration. The forest overstory is removed entirely or partially during conversion, and a new forest is planted on the spot. Transformative restoration involves gradually removing and replacing sections of the overstory”.

Completed.

  1. Check for the repetition.

“Beech nuts serve as forage for mammals at many trophic levels, and increased beech nut crops are associated with increased predatory animal populations” in Line-94,95 and “It serves as forage for mammals at many trophic levels, from myomorphic rodents to black bears (Ursus americanus), and increased beechnut crops are associated with increased predatory animals” in Lines 120-122.

The second instance to predatory animals has been removed in the sentence to reduce repetition.

  1. Cutting and removal of the existing vegetation may be necessary to remove both diseased and dying overstory  grandifoliatrees and non-resistant F. grandifolia regeneration, as well as other competing understory vegetation (e.g., invasive grasses, shrub specie” in Lines-382-385 and Removing existing vegetation may be necessary to cull both diseased and dying overstory F. grandifolia trees and non-resistant regeneration, as well as other competing understory vegetation in Lines-395-397.

The second sentence has been shortened to reduce repetition.

  1. L 401-405- Check and fix grammatical errors and paraphrase the sentence.

There is no grammatical error in this sentence.

  1. I suggest the authors to check and revise the manuscript for repetition of words, sentences etc.

The manuscript has been reviewed by multiple authors and additional reviewers.  We have reduced repetition where evident.  Given the complex nature of the review topic, multiple aspects of this species natural history may be important for different areas of sustainable management, thus some repetition is important for clarity in different areas of the document.

  1. Try to avoid repeating the complete genus name of the binomial name if it has already been mentioned  (Ref L. 61).

We believe this refers to the reference list. The species and binomial name is a part of the title of this reference item and would be incorrect to change.  Many references include binomials in the titles.

  1. In sub heading 2 the binomial  name is not in italics, similarly in L.80.

Fixed in response to Reviewer 3’s comments.

  1. Some paragraphs do not convey any meaning relative to the context. Ref L.340-345.

This is addressed in response to comment 13. The application of a restoration framework is common practice in restoration ecology and a critical component of the work presented. This critical work is introduced in the paragraph indicated.

  1. In figures, try to maintain the size of each component as the proportion of the size of the components in a figure make a lot of sense (Fig 2).

In figure 2, size of graphical elements was manipulated intentionally to represent the compositional structure one might expect to see in response to the treatments indicated (i.e., younger trees are smaller, older trees grow larger).  Other reviewers indicated approval of the figures and suggest they are appropriate and did not comment on issues related to them so we have not made changes suggested by this reviewer.

  1. In the reference section, some references lack their year of publication (Ref No. 22), and some references do not have their year in bold format (Ref. No. 34). Reference number 15 seems confusing. Reference number 34 has a 'pp' at the end without indicating the page number. Some references use pp before the page number, some do not.

References were formatted using Endnote and the style guidelines provided by the journal here https://www.mdpi.com/journal/sustainability/instructions.  References have been reviewed for minor edits in the revised document. Ref 15 had an extra space removed that had shifted part of it to a lower line. Bold formatting for years is indicated in the author guidelines only to be used for journal articles, not other types of references such as book chapters.  The optional pagination was moved to before numbers where appropriate (we believe this was an artifact of Endnote, but references have been carefully checked in this revision).

Reviewer 5 Report

Well structured and written. 

Author Response

Thank you for your review.

Round 2

Reviewer 3 Report

The authors respond well to all the suggested queries. Now the quality of the manuscript is much improved. Therefore recommended for publication in its present form.